# The Impact of Anatomical Predisposition and Mechanism of Trauma on Dislocation of the Patella: A Retrospective Analysis of 104 Cases

**DOI:** 10.3390/jpm13010084

**Published:** 2022-12-29

**Authors:** Ilona Schubert, Patrick Morris, Jörg Dickschas, Peter C. Strohm

**Affiliations:** 1Department of Orthopedics and Trauma Surgery, Sozialstiftung Bamberg, 96049 Bamberg, Germany; 2Department of Orthopedic and Trauma Surgery, Friedrich-Alexander-University of Erlangen, 91054 Erlangen, Germany; 3Department of Orthopedic and Trauma Surgery, Albert-Ludwig-University of Freiburg, 79085 Freiburg im Breisgau, Germany

**Keywords:** dislocation of the patella, predisposition, patellofemoral joint, patellofemoral dysbalance

## Abstract

Background: The aim of this study was to determine whether traumatic dislocation of the patella is provoked by the presence of predisposing factors and examine the role of the mechanism of injury. Methods: Cases diagnosed with dislocation of the patella and covered by the workers’ compensation program were identified and classified as traumatic based on insurance regulations. We examined predisposing factors (e.g., frontal axis, torsional deviation, trochlear dysplasia, patella alta) in case groups based on age at dislocation and trauma mechanism. Retrospective cohort study, level of evidence III. Results: Our sample size comprised 104 cases, consisting of 54 children and 50 adults. The most common mechanism of injury in children and adults was rotational trauma. Only 20% of the children and 21% of the adults exhibited no relevant predisposing factors. Group specifically, falls accounted for the highest number of cases exhibiting none of the defined anatomical predisposing factors. Children are more frequently affected by predisposition-related dislocations than adults. Conclusion: The proportion of predispositions is high. A fall, direct impact, or rotational trauma can be viewed as an adequate mechanism of trauma. For successful treatment, it is paramount to analyze the exact mechanism of the trauma and address any underlying predispositions.

## 1. Introduction

The instability of the patellofemoral joint has gained great attention from clinicians and researchers alike, as new revelations in biomechanics and treatment options surface. We often distinguish traumatic dislocation of the patella from habitual dislocation. For a traumatic dislocation to occur, there has to have been an adequate traumatic event, such as a direct medial impact to the patella. Habitual dislocations of the patella arise during the physiological motion of the knee joint and commonly occur repeatedly. This is due to inherent joint instability or secondary joint instability acquired through not sufficiently treated post-traumatic injuries. The patellofemoral joint is guided and correctly aligned by the interaction of active, passive, and static stabilizers. In the complete extension of the knee joint, the medial ligamentary complex acts as a passive stabilizer. At around 30° knee flexion, the patella seats itself in the trochlear groove and is supported, statically, by the osseous structures. During higher degrees of flexion, the quadriceps muscles act as active stabilizers. Anatomical abnormalities categorized as predispositions are those that promote a deviation from the optimum force vectors, thereby resulting in a lateralization of the patella or incongruence of the patella and trochlea. Known predisposing factors are valgus deformity, increased internal rotation of the femur, increased external rotation of the tibia, elevated TTTG-distance (tibial-tuberosity to trochlear groove distance), high-riding patella, or dysplasia of the trochlea or patella itself. Muscular hypotrophy and laxity of the ligamentary complex can also act as a predisposing factor. The evaluation of predisposing factors helps us to identify patients at risk for dislocation events as well as derive definitive treatment options in order to prevent the development of recurring dislocations and subsequent joint damage [1,2,3,4,5,6,7,8,9,10].

Patients who exhibit a high degree of predisposing factors have been shown to often suffer their first dislocation event as young adolescents [6,9,11]. However, the presence of a predisposition does not lead to a dislocation in every case. Depending on the severity of the predisposition, it can be difficult to distinguish whether the cause of a dislocation event is of traumatic nature or can be attributed to a predisposition. Therefore, the mechanism of trauma should be included in the process of patient assessment. A certain treatment regimen is then chosen based on the resulting injuries incurred by the dislocation event and the underlying risk factors for a dislocation to reoccur.

In the event of a first-time dislocation and lack of traumatic structural damage (e.g., flake fracture), non-surgical treatment options are usually chosen [6,12,13,14,15]. Surgical treatment is often indicated in the case of recurring dislocations of the patella after a thorough workup of predisposing factors has been performed. If the mechanism of trauma is inadequate and a relevant predisposition is present, surgical therapy should be considered even after a first-time dislocation event [6,14,16,17].

## 2. Aim of the Study

The aim of this study was to answer the question of whether the analysis of trauma mechanisms and certain predisposing factors in patellar dislocations can provide information about the main cause of a dislocation event. In analyzing the cause of patellar dislocations, we also wanted to examine the extent of differences between adult and pediatric patellar dislocations.

## 3. Methods

We included all patients who were treated in our clinic from January 2000 until April 2020 that were recorded with a diagnosis of “dislocation of the patella” on the standard form F1000 of the German statutory accident insurance. The cases that were not dislocations, after all, were excluded.

For data acquisition purposes we also utilized available digital imaging (X-ray, MRI, torsional-CT scans), as well as treatment documentation (out-patient documentation, discharge papers, surgical protocol).

Patients were grouped according to age with a cut-off at 18 years (smaller or equal to 18 and older than 18 years).

We evaluated general descriptive demographical aspects (age at trauma, gender), general information concerning the accident (mechanism of trauma, date of the traumatic event), and information concerning treatment (in-patient, total time of hospital stay, type of therapy, surgical technique chosen, time passed between trauma and surgical treatment). The data were obtained from the available form F1000 or the treatment documentation. For ease of systematical evaluation, we grouped the circumstances of the accidents in the subgroups direct impact, insufficient trauma, fall, rotational trauma, and unknown mechanism of trauma.

Furthermore, we evaluated anatomical aspects. The affected side, first-time dislocation or recurring, frontal axis, femoral and tibial torsion, TTTG value, height of the patella, shape of the trochlea according to Dejour [18], and lateral trochlear inclination angle. Frontal axis measurements were performed on full-length standing AP radiographs, and torsional angles were measured in CT scans according to the method of Waidelich [19]. Patella height was determined via the Insall-Salvati-Index [20] and Caton-Dechamps-Index [21]. Taking into consideration both methods of measurement, the cases were categorized as normal, normal to high-riding, normal to low-lying, high-riding, and low-lying. The shape of the trochlea was either classified as “normal” or categorized according to the Dejour subtypes A-D [18] by means of axial MRI scans. Additionally, we measured the lateral trochlear inclination, i.e., the resulting angle by drawing a tangent line along the lateral trochlear facet and the posterior condylar tangential line, also by means of axial MRI scans.

To analyze the relevance of predisposing factors, we determined the proportion of “knee-healthy” patients in each category with respect to the mechanism of trauma. Those labeled as “knee-healthy” exhibited measurement values that lie outside of the range of those patients who, in our clinic, are advised to undergo surgical treatment after having suffered a dislocated patella. Subjects were ruled out from the “knee-healthy” category if they exhibited a valgus deformity of greater or equal to 4 degrees, a femoral torsion of less or equal to -34 degrees, a tibial torsion of greater than or equal to 45 degrees, trochlear dysplasia Type D according to Dejour [18], a TTTG value of greater or equal to 16 mm or a high-riding patella. We excluded those subjects from our study group whose frontal axis, torsional values, and TTTG value could not be determined.

The descriptive data were recorded and analyzed with Microsoft Excel 365^®^. Descriptive analysis was performed primarily for both age groups, as well as in trauma category-associated subgroups. When numerical data was recorded, we specified the median and range [smallest value, largest value].

## 4. Results

Within the population of 18 years of age and younger, we were able to identify 54 events of a dislocated patella on 51 knees of 50 patients. Within the group older than 18 years of age we identified 50 events of a dislocated patella on 50 knees in 49 patients.

Due to inconsistent digital documentation practices, 11 cases from the adult group were derived from the years 2000 until 2010, and 39 cases from the years 2011 until 2020. In the pediatric subgroup 16 cases were derived from the years 2003 until 2010, and 38 cases from more recent years.

### 4.1. Adults

#### 4.1.1. Demographics

The population consisted of 36 males and 14 females. The median age at dislocation was 28 [19; 48] years. Cases consisted of 25 dislocations on the left knee and 25 dislocations of the patella on the right knee.

#### 4.1.2. Geometric Anatomical Data

Regarding the frontal axis, 20% presented a valgus configuration of 3 degrees [0°; 10°]. A similar number of patients (22%) presented with a varus configuration of 1 degree [0°; 4°]. In 22% of cases, we witnessed a zero-degree frontal axis, whereas no definitive conclusion on the frontal axis was reached in 36% of cases.

We found the median femoral torsion to be −24 degrees [−45°; −1°] and the tibial torsion to be 37 degrees [19°; 51°]. The original publication by Waidelich [19] defined the physiological range in patients older than 18 years of age as −20.4 ± 9.0 degrees femoral torsion and 33.1 ± 8.0 degrees tibial torsion. Therefore, our study group’s median values are within the physiological range. We saw an increased tibial torsion (threshold value −24°) in 24% of cases and an increased tibial torsion (threshold value 35°) in 28% of cases.

The median TTTG value was 18 mm [8 mm; 35 mm] and was greater than 15 mm in 54% of cases. Surgical treatment is commonly recommended after a dislocation event and a present TTTG value of greater than 15 mm [14,22].

Examining subjects for a high-riding patella, we calculated a median Insall-Salvati-Index of 1.2 [0.7; 1.5] and median Caton-Dechamps-Index of 1.2 [0.8; 1.5]. A physiological range for the Insall-Salvati-Index is defined as 0.8–1.2 [20], and for the Caton-Dechamps-Index 0.6–1.2 [21]. We found 16% to be categorized as a normal to high-riding patella and 28% as high-riding. One case was found to be normal to low-lying, whereas no cases were low-lying and 46% were defined to be normal.

The trochlear shape was defined as physiological in 42% of cases, while 22% of cases could not be classified due to a lack of data. A morphological deformity (type B 56%, type C 33%, type D 11%) was displayed in 36% of cases Lateral trochlear inclination serves as a tool to quantify the slope of the trochlea and is also seen as a predisposing factor for the dislocation of the patella. The pathological threshold value is specified at 11 degrees [22,23]. The median in the adult group was 16 degrees [7°; 24°]. Values of less than 11 degrees were shown in 16% of cases.

#### 4.1.3. Effects of Trauma

The adult group consisted of 33 first-time and 17 recurrent dislocations. Injury to the medial patellofemoral ligament (MPFL) resulted in a complete rupture in 72%, and partial rupture or bony avulsion in 14% of cases. Due to a lack of data, no conclusion on MPFL injury could be drawn in 14% of cases. Flake fractures were present in 34% of cases. Bone bruise of the lateral femoral condyle was seen in 84% of cases. In 16%, no statement could be made regarding the aforementioned injury, while the absence of this pathognomonic phenomenon was not described in any case.

#### 4.1.4. Treatment

In 32% of cases, non-surgical treatment was initiated, of which 10% ultimately ended up undergoing surgical treatment. Surgical treatment was described in a total of 62% of cases. In 16%, it was not possible to assign a treatment option due to a lack of data. Of the first-time dislocations, 52% were treated surgically and 33% non-surgically (no assignment possible in 15% of cases); of the recurrent dislocations, 53% were treated surgically and 29% non-surgically (no assignment possible in 18% of cases). The median duration from the time the injury was incurred to surgical treatment was 12 [1; 1100] days. In subgroup analysis, the median time to surgery, including those who primarily received non-surgical treatment, was eight [1; 137] days for primary dislocations and 24 [1; 1100] days for recurrent dislocations. The median duration of in-patient stay was three [0; 20] days, and 16 cases were treated as outpatients.

Regarding the chosen surgical treatment, reconstruction of the medial patellofemoral ligament was the most common (55%). In the case of osseous deformity correction, the following were described: 19% of cases received an osteotomy of the tibial tuberosity, a femoral or tibial osteotomy was performed in 10%, and a trochleoplasty was not documented in any case. Reattachment of a flake fracture or a bony avulsion of the MPFL was performed in 19% of cases.

### 4.2. Children and Adolescents

#### 4.2.1. Demographics

The population consisted of 23 females and 31 males. The median age was 14 [9; 18] years. Cases consisted of 26 dislocations on the left and 28 dislocations of the patella on the right knee.

#### 4.2.2. Geometric Anatomical Data

Regarding the frontal axis, 39% of cases showed a valgus configuration of 3 degrees [2°; 8°]. 12% of cases presented with a median varus configuration of 2 degrees [1°; 4°], 12% of cases had a straight leg axis and in 37% no conclusive statement could be made regarding the frontal axis.

We found the median femoral torsion to be −26 degrees [−51°; −12°] and median tibial torsion to be 32 degrees [17°; 48°]. We saw an increased femoral internal torsion (threshold value −24°) in 25% of cases and increased tibial external torsion (threshold value 35°) in 18% of cases. The median TTTG value was 17 mm [8 mm; 24 mm] and exceeded 15 mm in 49% of cases.

Examining patellar height, the Insall-Salvati index yielded a value of 1.3 [0.8; 2], and the Caton-Dechamps index, a value of 1.3 [0.5; 1.7]. These indices demonstrate good reliability for children despite ongoing growth [24,25]. Again, taking into consideration both methods of measurement, we found 22% to be categorized as normal to high-riding patella and 41% as high-riding. Only one case was found to be normal to low-lying and no case was low-lying. A normal height was shown in 24% of cases.

The trochlea showed a normal configuration in 20% of cases, in 23% no trochlea shape could be determined due to a lack of data. In 57% of cases a pathological trochlea configuration was found (type B 38%, type C 41%, type D 21%). The median lateral trochlear inclination angle was 12 degrees [4°; 21°], and 33% of cases showed values less than 11 degrees.

#### 4.2.3. Effects of Trauma

The adolescent group consisted of 42 first-time dislocations and 12 recurrent dislocations. Complete rupture of the medial patellofemoral ligament occurred in 67% of cases and partial damage or elongation in 20%. A flake fracture or bony avulsion was described in 41% of cases. Bone bruise of the lateral femoral condyle was seen in 76% of cases. In 15%, no statement could be made regarding the aforementioned injury due to lack of data, while the absence of this pathognomonic phenomenon was described in 9%.

#### 4.2.4. Treatment

In the pediatric group, 26% of the patella dislocations were treated non-surgically and 69% were treated surgically, in 5% no allocation to primary treatment groups could be made due to a lack of data. Considering first-time dislocation events, surgery was performed in 67% of cases, and non-surgical treatment was chosen in 29% (no assignment to treatment groups possible in 4% of cases); of the recurrent dislocations, 75% of cases were treated surgically and 17% of cases were treated non-surgically (no assignment possible in 8%). The median duration from the time the injury was incurred to surgical treatment was 12 [1; 214] days, in the subgroup of first-time dislocations, it was only eight [1; 34] days, while in the subgroup of recurring dislocations it was 15 [5; 214] days. The median duration of in-patient stay was three [0; 21] days, of which 12 cases only required outpatient treatment and thus accounted for a stay of zero days.

Regarding the surgical treatment options, addressing the medial ligamentary complex was the most common in 65% of cases. In 30% of cases among older adolescents, an autologous augmentation of the MPFL was performed. In the case of osseous deformity correction, the following were described: osteotomy of the tibial tuberosity in 22%, growth control as temporary hemiepiphysiodesis in 16%, reattachment of flake fractures or bony avulsions in 24%. Trochleoplasty was performed in one case of a 16-year-old male patient and recommended in one other case.

### 4.3. Categories of Trauma Mechanisms

When considering different mechanisms of trauma that cause an event of a dislocated patella, some seem more plausible than others. In order to draw conclusions on the extent to which “traumatic” patellar dislocations are triggered by predisposing factors, the raw data was broken down by accident category and the proportion of “healthy knees” in each category was determined (see Table 1, and Figure 1). Within the entire adult cohort, the proportion of “knee-healthy” patients was 21%; within the pediatric cohort, it was 20%. If the cutoff TTTG value is raised from ≥16 mm to ≥19 mm, the ratio increases to 28% in the adult group and to 24% in the pediatric group.

#### 4.3.1. Rotational Trauma

Rotational trauma was the most common trauma entity, accounting for 42% in the pediatric population (first-time dislocation in 78%; recurrent dislocation in 22%) and 52% in the adult cohort (first-time dislocation in 73%; recurrent dislocation in 27%). The analysis for predisposing factors in this trauma category is shown in Table 1.

#### 4.3.2. Direct Impact

Direct impact to the patella caused a dislocation event in 10% within the adult cohort and in 19% within the pediatric cohort (first-time dislocation in 80% and recurrent dislocation in 20% of cases in both age groups). In this trauma category, 20% of adults and 13% of children were “knee-healthy”. In the adult population, the trochlear shape was defined as normal in 80%, while no assignment regarding the trochlea shape was possible in 20%. See Table 1.

#### 4.3.3. Fall

13% of children and 8% of adults were assigned to this category. Falls resulted in a first-time dislocation event in 100% of both age groups. There were a significant number of cases within the adult group where a numerical value for the frontal axis and torsional values could not be determined due to a lack of digital imaging studies and/or lack of documentation. Therefore, we excluded the calculation of median torsional values as well as the percentage of valgus deformities in order to prevent bias. See Table 1.

#### 4.3.4. Insufficient Trauma

15% of pediatric dislocations and 20% of adult dislocations were attributed to this category. This trauma category showed the highest rate of recurrent dislocations (50% of cases within the children cohort and 90% of cases within the adult cohort). The ratio of patellofemoral abnormalities was high and the proportion of “healthy knees” was lowest, at 13% in adult cases and 5% in pediatric cases. See Table 1.

## 5. Discussion

Rates for recurrent events after a first-time dislocation event vary among different authors [6,17]. The risk of suffering another dislocation is higher after having suffered a prior dislocation, especially when the first-time dislocation occurred at a young age [5,14,26]. The goal of successful treatment, on the one hand, is the adequate treatment of the acute trauma effects such as osteochondral damage, on the other hand, the (re-)establishment of stable joint integrity. The consequences of recurrent dislocations are far-reaching, ranging from functional limitations attributed to instability, persistent pain, and reduced stress capacity of the joint, to morphological damage such as cartilage-bone pathologies with increased risk of degenerative joint disease and reduced quality of life [6,12,27].

Traumatic first-time dislocations are usually treated non-surgically unless relevant osteochondral flakes are present [6,12,13,14,15]. However, when is the cause of a dislocation event considered traumatic? If adequate trauma cannot be assumed, the cause must be sought in potential physical abnormalities. A concept of treatment can only be causally determined after a thorough evaluation of all potential underlying factors influencing the dislocation. Germany’s statutory accident insurance will cover an injury, including those with pre-existing conditions, if a certain traumatic event can, beyond a reasonable doubt, be legally accepted as the cause. Only 21% of adult knees and 20% of children’s knees in our cohort showed an absence of relevant pathological deviations. Overall, our study population of traumatic dislocations showed a large proportion of predisposing factors.

A valgus configuration was a frequent predisposing factor with 20% in the adult cohort and almost twice as many in the pediatric cohort. The median values, however, did not exceed our correction limit of four degrees. There are currently no clear recommendations on the degree of axis deviation at which axis correction should be performed for patella maltracking [28], but frontal axis deviation seems to only play a minor role in traumatic dislocations.

Median torsional values also only deviated moderately from the norm for both cohorts. Considering the above-mentioned threshold values, however, clear pathologies were found in individual cases. Both groups had a similar number of cases that exceeded the threshold value for femoral torsion, while the adult population had a higher number of cases above the threshold for tibial torsion than the pediatric cohort. Thus, torsional deformities in childhood seem to play only a minor role [29]. In adults, a torsional osteotomy is recommended with a symptomatic deviation of approximately 10 degrees or more [4,28]. Dickschas et al. [4] showed that about 12% of patella instabilities are caused by torsional deformities.

With a median TTTG value of 17 mm within the pediatric and 18 mm within the adult cohort, both were slightly elevated, while both groups showed a TTTG value above the correctional limit of 15 mm [14,22] in approximately half of the cases. The risk factor “high-riding patella” also showed increased relevance, with an incidence of just under 30% in adult cases and just over 40% in pediatric cases. Osteotomies of the tibial tuberosity were performed less frequently than would be expected based on the proportion of underlying pathology.

Consistent with the increased number of cases that demonstrated trochlear dysplasia within the pediatric cohort (57%), the lateral trochlear inclination angle was less than 11 degrees in approximately one-third of cases. Unexpectedly, the adult cohort exhibited some form of trochlear abnormality in 36% of cases. In severe cases of trochlea dysplasia, a surgical re-shaping (trochleoplasty) may be indicated. Trochleoplasty may influence other predisposing factors, as patellar tilt and patellar height are influenced by a dysplastic trochlear geometry, and reconfiguration of the trochlea influences the TTTG value [28]. Open growth plates are usually considered a contraindication for this procedure, in which case, one should resort to surgical procedures addressing the balance of associated soft-tissue structures.

Schmeling et al. [28] describe a classification system by naming five types of patellofemoral instability, which may assist clinicians in choosing an appropriate treatment regimen. Type 1 describes a traumatic patellar dislocation without maltracking or instability with a low risk of re-dislocation. Type 2 describes patellar instability without maltracking with a high risk of re-dislocation. Type 3 is defined as patellar instability with maltracking. Type 4 describes high-grade instability with loss of patellar guidance due to high-grade trochlear dysplasia, while type 5 is characterized by maltracking without instability. Types 1 and 2 can be easily interpreted and understood as traumatic events. However, the present study population demonstrated an increased incidence of predisposing factors hinting at pre-existing patellar maltracking, making it a necessity to evaluate the mechanism of trauma.

Mechanisms of trauma such as stepping out of a vehicle or standing up from a seated position correspond to an everyday movement and are not viewed as adequate mechanisms of trauma for a patella dislocation to occur and were therefore grouped within the category “insufficient trauma”. Accordingly, this subgroup showed a low rate of “healthy knees” in both age groups. The children’s cohort in particular was characterized by a high proportion of trochlear dysplasia and a high-riding patella. The question arises why the proportion of this trauma category is also high in the adult cohort, as dislocations, given a relevant predisposition, would be expected to have also occurred at a younger age. This can be explained by the high proportion of recurrent dislocations within the adult cohort within the “inadequate trauma” category. Thus, these are not to be considered as traumatic. According to the allocation to type 3 or 4 of patellofemoral instability [28], one should aim for a causal surgical therapy.

Direct impact to the medial aspect of the knee, on the other hand, can plausibly cause a traumatic dislocation [8,30]. This theory is supported by the low proportion of recurrent dislocations and the correspondingly high proportion of first-time dislocations, as well as by the rather low proportion of underlying predisposing factors in the adult group. Comparatively, a pathological trochlear shape, patellar height, and lateral trochlear inclination angle could be witnessed more often in the pediatric cohort. There is also a lower proportion of “healthy knees” in the pediatric age group. This shows that clinicians should not be misled by the plausibility of certain mechanisms of trauma and dismiss other variables, especially in children. It remains important to consider underlying predisposing factors before initiating a certain treatment option.

The pediatric subgroup in the trauma category “rotational trauma” also showed an increase in pathological values regarding predisposing factors compared to the adult subgroup. Within the latter, however, the older age of the injured policyholders was noticeable, further supporting the trauma thesis.

With 100% first-time dislocations and the highest proportion of “healthy knees” in both age groups (60% in the pediatric cohort, 33% in the adult cohort), the trauma category “fall” represents a relevant and major cause for traumatic dislocation of the patella.

Ultimately, the surgical treatment chosen should be based on the presence of relevant predisposing factors and the resulting injuries incurred due to the dislocation event itself. In accordance with the recommendation for type 1, i.e., non-surgical treatment as long as there is no relevant osteochondral damage [28], non-surgical treatment was initiated in 22% of adult cases and 26% of pediatric cases, corresponding to the proportions of “healthy knees”. In type 2, surgical treatment addressing the medial soft tissue is recommended [28], while MPFL reconstruction has been shown to be superior to duplicating the retinaculum in adults [31]. In children, MPFL reconstruction is still considered critical due to the insertion point being located at the distal femoral growth plate [6]. Although there are studies that show a reduced risk of re-dislocation after surgical stabilization after having suffered a traumatic first-time dislocation with subsequent instability, long-term satisfaction in clinical scores is not significantly higher compared to non-surgical treatment [13,15,16]. In the present study cohort, surgically addressing the medial ligamentary complex was the most common procedure performed in children and adults, in line with the above-mentioned recommendation.

Contrary to one’s assumption on the treatment of mere traumatic dislocations of the patella, our study population underwent correctional osteotomies by means of tibial tuberosity osteotomy, torsional and frontal axis corrections, or trochleaplasty, as recommended for maltracking instability (types 3 and 4 [28]). Within these cases, a sufficient mechanism of trauma is deemed questionable.

## 6. Conclusions

Dislocations of the Patella are less often of merely traumatic nature, as they are triggered by geometric malformities of the knee. Children are affected by predisposition-related dislocations more frequently than adults. Depending on the mechanism of trauma, the proposition of non-surgical treatment after a first-time dislocation event without concomitant osteochondral injury should be abandoned. Falls are an adequate mechanism of trauma. In cases of direct impact and rotational trauma, relevant trochlear dysplasia should be ruled out before trauma adequacy is determined. Especially in the cases of the insufficient mechanism of trauma, an analysis of the known predisposing factors should be performed before the initiation of a causal therapy, even in the case of a first-time dislocation event. In some cases, a definitive causal therapy can only be performed after the growth plates have closed.

## Figures and Tables

**Figure 1 jpm-13-00084-f001:**
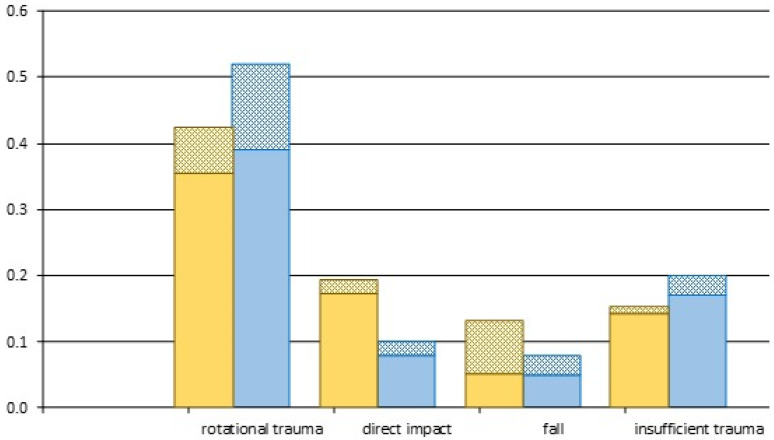
The distribution in terms of trauma categories: Beige columns represent the pediatric study population, blue columns the adult population. In each column, the proportion of “healthy knees” is hatched.

**Table 1 jpm-13-00084-t001:** The proportion of different predisposing factors and “healthy knees” in the individual trauma categories and age groups.

	Rotational Trauma		Direct Impact		Fall		Insufficient Trauma	
	Pediatric Group	Adult Group	Pediatric Group	Adult Group	Pediatric Group	Adult Group	Pediatric Group	Adult Group
median age (years)	14 [9; 18]	28 [19; 45]	14 [11; 17]	23 [19; 30]	14 [10; 18]	44 [32; 48]	17 [13; 18]	26 [20; 40]
proportion of valgus deviation (%)	32	23	33	40	57	/	29	10
median tigh torsion (°)	−29 [−44; −12]	−28 [−45; −1]	−27 [−51; −22]	−24 [−31; −20]	−22 [−25; −19]	/	−24 [−46; −13]	−21 [−33; −18]
median lower leg torsion (°)	30 [23; 48]	35 [28; 40]	36 [17; 46]	38 [27; 51]	23 [17; 38]	/	33 [28; 40]	39 [19; 42]
TTTG value (mm)	17 [9; 24]	18 [10; 27]	17 [8; 21]	16 [11; 35]	14 [8; 24]	19 [17; 22]	16 [10; 23]	18 [8; 22]
median lateral trochlear inclination (°)	14 [8; 21]	16 [7; 20]	13 [9; 18]	19 [17; 24]	17 [4; 20]	11 [9; 15]	9 [7; 16]	18 [9; 24]
ratio of trochlea dysplasia	50	38	78	0	43	75	50	20
patella height: proportion of normal to high-riding and high-riding (%)	64	42	89	60	43	25	57	70
proportion of “healthy knees” (%)	16	25	13	20	60	33	5	13

## Data Availability

Data is contained within the article.

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
