# Peer review of "The Impact of Anatomical Predisposition and Mechanism of Trauma on Dislocation of the Patella: A Retrospective Analysis of 104 Cases"

_jpm, 2022, doi:10.3390/jpm13010084_

Round 1

Reviewer 1 Report

A very well written paper. Good English and Grammar.

Introduction - good and extensive

Methods: Inclusions well described. Very descriptive

Results: Detailed results. Very well done and written.

Discussion: Other studies included and papers described

Conclusion: Good conclusion 

Overall a very well written paper, I do not think I can improve much on this paper. I would accept for publication.

Author Response

Thanks to the reviewer!

Reviewer 2 Report

The subject is difficult and  controversial. 

It is a retrospective study with all associated shortcomings (Small cohort) .

I does present a sound theoretical base for evaluation and treatment of patellar dislocations.

The high surgery rate for first time dislocations may be viewed as "novel" by some of the readership in the absence of long term follow up.

Excellent "food for thought" and incentive for prospective/randomized studies. In my opinion, best feature of the article.

Author Response

Thanks to the reviewer!

Reviewer 3 Report

Major comments

1. Exclusion criteria need to be mentioned. Inclusion criteria should also well defined

2. outcome data collected: methodology adopted to collect biomechanics or morphometric data. were they copied from insurance data or collected by questionnaire or else  not clear for readers

3. Appropriate statistical analysis for claim there hypothesis. Simple prevalence measurement could not much helpful

Minor

English corrections required.

Comparison of literature could be avoided in result section, better to placed in discussion section

Details comments in attached document

Round 2

Reviewer 3 Report

Congratulations